# Effect of El Niño Southern Oscillation cycle on the potential distribution of cutaneous leishmaniasis vector species in Colombia

**Mariano Altamiranda-Saavedra[1]ᵒ, Juan David Gutiérrez[2]ᵒ, Astrid Araque[3], Juan David Valencia-Mazo[4], Reinaldo Gutiérrez[5], Ruth A. Martínez-Vega[6]ᵒ \***

**1** Grupos de investigación COMAEFI y SIAFYS, Politécnico Colombiano Jaime Isaza Cadavid, Medellín, Antioquia, Colombia, **2** Grupo Ambiental de Investigación Aplicada-GAIA, Facultad de Ingeniería, Universidad de Santander, Bucaramanga, Santander, Colombia, **3** Laboratorio de Salud Pública de Norte de Santander, Instituto Departamental de Salud, Cúcuta, Norte de Santander, Colombia, **4** Grupo Mastozoología, Instituto de Biología, Universidad de Antioquia, Medellín, Antioquia, Colombia, **5** Grupo de Investigación GIEPATI, Universidad de Pamplona, Pamplona, Norte de Santander, Colombia, **6** Grupo de Investigación Salud-Comunid-UDES, Programa de Medicina, Universidad de Santander, Bucaramanga, Santander, Colombia

ᵒ These authors contributed equally to this work.
\* rutharam@yahoo.com

**Data Availability Statement:** All relevant data are within the manuscript and its Supporting Information files.

## Abstract

Local anomalies in rainfall and temperature induced by El Niño and La Niña episodes could change the structure of the vector community. We aimed to estimate the effect of the El Niño–La Niña cycle in the potential distribution of cutaneous leishmaniasis (CL) vector species in Colombia and to compare the richness of the vectors with the occurrence of CL in the state of Norte de Santander. The potential distributions of four species were modeled using a MaxEnt algorithm for the following episodes: La Niña 2010–2011, Neutral 2012–2015 and El Niño 2015–2016. The relationship between the potential richness of the vectors and the occurrence of CL in Norte de Santander was evaluated with a log-binomial regression model. During the El Niño 2015–2016 episode, *Lutzomyia ovallesi* and *Lutzomyia panamensis* increased their distribution into environmentally suitable areas, and three vector species (*Lutzomyia gomezi*, *Lutzomyia ovallesi* and *Lutzomyia panamensis*) showed increases in the range of their altitudinal distribution. During the La Niña 2010–2011 episode, a reduction was observed in the area suitable for occupation by *Lutzomyia gomezi* and *Lutzomyia spinicrassa*. During the El Niño 2015–2016 episode, the occurrence of at least one CL case was related to a higher percentage of rural localities showing a richness of vectors = 4. The anomalies in rainfall and temperature induced by the episodes produced changes in the potential distribution of CL vectors in Colombia. In Norte de Santander, during Neutral 2012–2015 and El Niño 2015–2016 episodes, a higher probability of at least one CL case was related to a higher percentage of areas with a greater richness of vectors. The results help clarify the effect of the El Niño–La Niña cycle in the dynamics of CL in Colombia and emphasize the need to monitor climate variability to improve the prediction of new cases.

**Funding:** The authors received no specific funding for this work.

**Competing interests:** The authors have declared that no competing interests exist.

## Author summary

The cutaneous leishmaniasis is a disease transmitted by insects. The incidence of cutaneous leishmaniasis has increased in Colombia and the state of Norte de Santander is one of the Colombian states where cutaneous leishmaniasis transmission is high. Local changes in rainfall and temperature induced by El Niño and La Niña episodes could change the distribution of the vector. A database of published records and field collections of four vectors of cutaneous leishmaniasis in Colombia was compiled. Also, a database with cases of cutaneous leishmaniasis from Norte de Santander was obtained. Maps of potential distribution in Colombia of the four vectors during the La Niña 2010–2011, Neutral 2012–2015 and El Niño 2015–2016 episodes were elaborated. During the El Niño 2015–2016 episode, two vector species increased their distribution into environmentally suitable areas, and three vector species showed increases in the range of their altitudinal distribution. During the La Niña 2010–2011 episode, a reduction was observed in the area suitable for occupation by two vectors. During the El Niño 2015–2016 episode, the occurrence of at least one cutaneous leishmaniasis case was related to a higher percentage of area with a predicted distribution of four vectors.

## Introduction

Cutaneous leishmaniasis (CL) is a vector-borne disease caused by flagellate protozoa of the genus *Leishmania* and is transmitted to humans by the bite of *Phlebotominae* (Diptera: *Psychodidae*) insects. This vector-borne disease is present in most tropical regions. Globally, approximately 200,000 new cases occur per year; however, considering the underreporting, it has been estimated that approximately 0.7 to 1.2 million CL cases occur each year [1]. More than 70% of all CL cases worldwide occur in only 10 countries, including Colombia [2]. Between 2001 and 2005, the incidence increased by 4.4-fold in Colombia [3]. Norte de Santander is one of the Colombian states where CL transmission foci are present. Norte de Santander accounted for 11.3% of all CL cases in Colombia in 2015 and 2016, although it contains only 1.5% of the country's population [4]. Thus, this region is a relevant place to focus in the effect of the El Niño-Southern Oscillation cycle on the potential distribution of CL vector species and the relationship with the occurrence of cases during the episodes of El Niño and La Niña.

The relationship between the occurrence of CL cases and environmental factors has been studied to identify areas of higher risk. Land cover has been evaluated in Colombian municipalities to predict municipalities with at least one case of CL [5] or to identify the dependence of municipality incidence on land use, climate, elevation and population density in the Andean zone [6].

One hundred sixty-three species of *Phlebotominae* have been recorded in Colombia, but only 14 species have been identified as vectors of leishmaniasis (cutaneous, visceral or mucocutaneous) [7]. The distribution of sand flies of medical importance in Colombia corresponds predominantly to disturbed areas, where the original land cover is missing, which increases the potential for domiciliation by the sand flies [7]. CL vectors require microhabitats with high humidity, such as moist cracks near water sources, which are favorable natural breeding sites [8]. The occurrence of CL vectors has been associated with climatic factors, and changes in temperature and rainfall can affect the life cycle of vectors. For example, periods with high temperatures and no rain deteriorate microhabitats conducive to the development of the larval stages; likewise, extremely rainy periods that flood the soil can affect the survival of the vectors [9–11].

The El Niño-Southern Oscillation (ENSO) is characterized by unusual temperatures in the equatorial Pacific Ocean. This anomaly has the power to change the global atmospheric circulation and climate patterns. In Colombia, the warm phase of the oscillation (El Niño) leads to periods with high temperature and decreased rainfall, river flow, and soil moisture, as well as to low atmospheric humidity in the Andean, Caribe and Pacific regions. Meanwhile, the cold phase (La Niña) corresponds to low temperature; increased rainfall, river flow, and soil moisture and high atmospheric humidity in the same regions of the country.

Evidence exists of the impact of El Niño on the occurrence of cases of leishmaniasis in some regions of Colombia [9,10]. The predicted geographic distributions of the vectors of CL and the effects of climate change in North and Central America [12] and Brazil [13] have been previously analyzed.

Local anomalies in rainfall and temperature induced by the El Niño and La Niña episodes can cause changes in the structure of the community of vectors due to hydrological and physiographic factors such as the percentage of shade, soil moisture, depth of the forest leaf-litter and erosion that modifies the soil movement and stability and thus can also influence vector species distribution on the forest floor [14].

In addition to the changes in the structure of the community of vectors, the episodes of El Niño and La Niña can induce changes in the population dynamics of vectors. For example, in Panama, some vector species peak in density during the wet season, including *Lutzomyia panamensis* and another wet-adapted species, just as *L. gomezi* decreased significantly during the dry season [14]. Other studies on the patterns of adult emergence have concluded that CL vector species are mainly seasonal [15].

The most common vector species of CL in Norte de Santander are *L. spinicrassa* [7,16], *L. gomezi* [7,17,18], *L. ovallesi* and *L. panamensis* [7]. Previous studies have reported *Leishmania panamensis* as the most frequent causative agent of this clinical form [19]. Other common parasites in this region of Colombia are *Leishmania brazilensis* [20–22], *Leishmania mexicana* and *Leishmania amazonensis* [19].

In our knowledge, not exist a previous study that simultaneously appraise; a) the effect of the ENSO cycle on the potential distribution of the vectors of CL, and b) evaluate the effect of the changes in the potential distribution of vectors on the occurrence of cases of CL. Therefore, the aim of this study was to estimate the changes occurring in the potential distribution of four vector species of CL (*L. gomezi*, *L. ovallesi*, *L. panamensis* and *L. spinicrassa*) because of the changes in temperature and rainfall induced by the ENSO cycle in Colombia and to analyze if these changes in the predicted geographic distribution of the vectors are related to the occurrence of CL cases in the state of Norte de Santander.

## Materials and methods

### Records of vector presence

A database of published records of the presence of *L. gomezi*, *L. ovallesi*, *L. panamensis* and *L. spinicrassa* in Colombia from 1967 to 2016 was compiled from specialized literature. The literature search was conducted in the PubMed and SciELO databases using the terms "cutaneous" and "leishmaniasis" and "Colombia" and "*Lutzomyia*" (September 2017). The references for the presence records are cited in the supporting information (S1 Appendix). The literature records included some collection sites identified by coordinates and other sites with stated localities but no specific coordinates. Additionally, we included the field collection data of the Institute of Health of Norte de Santander. In the case of localities from the Institute of Health of Norte de Santander and published localities without specific coordinates, we assumed that the coordinates of the centroid of the rural locality were the collection point (S1 Appendix).

For theses localities, the value of uncertainty for the coordinates was calculated as the polygon perimeter size of the rural locality using ArcGis 10.3 (S1 Fig).

## Case data and case definition

Data on CL cases in humans were supplied by the Institute of Health of Norte de Santander from January 2007 to December 2016. We excluded from our analysis cases with incomplete or non-existent information about locality or notification date. A case of CL was defined as a patient who had skin lesions and met three or more of the following criteria: 1) the patient had no history of trauma, 2) the lesion evolved over more than two weeks, 3) the patient had a round or oval ulcer with raised edges, 4) satellite lesions were present, 5) nodular lesions were present, or 6) localized adenopathy was present. All cases were confirmed by detection of amastigotes of the genus *Leishmania* in smears or biopsies. There was no identification of *Leishmania* species. The place of residence was used to georeference each case because information on the possible place of transmission was not available. Additionally, cases before 2007 were not included because the data were grouped by state.

## Climatic information

WorldClim global climate data variables with spatial resolution of 30 arcseconds [23] were employed because this spatial resolution was compatible with the extension of rural localities in Norte de Santander, where we evaluated the relationship between the potential richness of the vectors and CL occurrence by episode of El Niño and La Niña (see section 2.5). Pearson's correlation analysis was performed to reduce the collinearity among the environmental layers with the "SDMtoolbox" tool in ArcGIS 10.3, and the variables with correlation value $> |0.8|$ were removed. Additionally, we use the Jackknife option in the software MaxEnt to identify variables that do not contribute significantly to the robustness of the models. Finally, for all models, we used the WorldClim variables BIO1 = Annual Mean Temperature and BIO12 = Annual Precipitation, with spatial resolution = 30 arcseconds and temporal range = 1950–2000. In addition to the bioclimatic variables, was included the land cover layer provided by the Global Land Cover Facility [24], reclassified to include the coverages present in Colombia (bodies of water, forest, mixed pasture, scrub, pastures, crops, bare ground, urban area and buildings). This layer was used because the heterogeneity in the risk of disease transmission results from spatial heterogeneity in both land cover and land use [25].

The study data reflect the period from January 2007 to December 2016. The time span of the episodes of the ENSO cycle between 2007 and 2016 was defined according to the values of the ONI index as provided in the National Oceanic and Atmospheric Administration database [26] (S1 Table). Temperature and rainfall data for Colombia were provided by the Institute of Hydrology, Meteorology and Environmental Studies (IDEAM), and 1,998 pluviometric and 519 temperature stations were analyzed.

Climate information from meteorological stations administered by IDEAM was used to estimate the changes in temperature and rainfall during the episodes of La Niña 2010–2011, Neutral 2012–2015 and El Niño 2015–2016 because these episodes corresponded to the most extreme events with the longest duration of their type during the last decade, and the neutral episode was the most extensive (S1 Table). We excluded the episode of La Niña 2007–2008 because according to the Institute of Health of Norte de Santander the 2007 report of CL cases was inaccurately documented by the surveillance system (mainly in rural areas); this phenomenon could correspond to a conditioning period in the healthcare institutions.

The anomalies were estimated based on the methodology of Montealegre (2014) [27]. For temperature data, an index of the difference between the accumulated value of temperature

during the time span and the historic average value of that period was calculated (Eq 1). The procedure for rainfall requires the relationship between the cumulative value of rainfall during the time span and the historic average value of that period (Eq 2).

$$A_t = \sum_{jM_b}^{jM_e} \frac{t_{ij}}{n} - \sum_{jM_b}^{jM_e} \frac{t_j}{n} \tag{1}$$

In the above equation, $At$ is the anomaly index of the temperature; $Mb$ and $Me$ are the beginning and ending months of the time span, respectively; $tij$ is the temperature in the time span $j$ of year $i$; $tj$ is the multiyear average temperature for time span $j$; and $n$ is the number of months of the time span, which is estimated as the difference between $Mb$ and $Me$.

A value of $At$ = 1.5 represents an increase in temperature of 1.5˚C, and a value of $At$ = -0.5 represents a decrease of 0.5˚C, with respect to the multiyear average temperature.

$$A_r = \sum_{jM_b}^{jM_e} r_{ij} \div \sum_{jM_b}^{jM_e} r_j \tag{2}$$

In the above equation, $Ar$ is the anomaly index of rainfall; $Mb$ and $Me$ are the beginning and ending months of the time span, respectively; $rij$ is the rainfall in the time span $j$ of year $i$; and $rj$ is the multiyear average rainfall for time span $j$.

A value of $Ar$ = 1.5 represents a rainfall excess of 50%, and a value of $Ar$ = 0.5 represents a rainfall shortage of 50%.

$At$ and $Ar$ were calculated for each climate station. We used empirical Bayesian kriging to interpolate the spatial data $At$ and $Ar$ to 30 arcseconds in ArcGIS software v 10.3. To obtain the values of temperature and precipitation during La Niña 2010–2011 and El Niño 2015–2016, we developed a raster layer with the arithmetic sum of $At$ and Annual Mean Temperature (BIO1) and $Ar$ and Annual Precipitation (BIO12).

## Potential distribution models

We hypothesized the historically accessible area (M area) to sample the background data [28]. A digital elevation model was used to extract the altitudinal distribution range for the records used in the models and the M area per species was designed using a buffer with the altitudinal range; *L. gomezi* = 0–2,600 m.a.s.l., *L. ovallesi* = 0–1,900 m.a.s.l, *L. panamensis* = 0–2,200 m.a.s.l. and *L. spinicrassa* = 450–3,200 m.a.s.l. These buffers were used as the model calibration areas. The program MaxEnt 3.3.3k [29] was used to estimate environmental suitability in these analyses. Different settings were tested using the ENMeval package of the R program to establish the optimal parametrization of the suitability estimates in the calibration region [30]. We selected the models with the lowest delta corrected Akaike Information Criterion score. The regularization multiplier and feature combination for *L. gomezi* were 1.5- LQHPT, for *L. ovallesi* 0.5-LQ, for *L. panamesis* 2.0-LQH and for *L. spinicrassa* 1.5- LQHP.

For each CL vector, three models of potential distribution were elaborated: Model 1 was a neutral model for the Neutral episode 2012–2015 with the variables of rainfall, temperature (both from WorldClim) and land cover; Model 2 was a model with temperature and rainfall during the La Niña episode of 2010–2011 and land cover; and Model 3 was a model with temperature and rainfall during the El Niño episode of 2015–2016 and land cover.

The model of the potential distribution of CL vectors was first calibrated and evaluated with neutral conditions (Neutral episode of 2012–2015) of Temperature (BIO1) and Rainfall (BIO12) (Model 1) and transferred to the La Niña episode of 2010–2011 (Model 2) and the El Niño episode of 2015–2016 (Model 3). To provide a further check on the reliability of our

model transfers, we calculated the mobility-oriented parity (MOP) metric with 5%, this analysis allows to evaluate the climatic similarity between the calibration area (M) and different climatic conditions [28].

The models were evaluated for the Neutral episode of 2012–2015 in the same calibration area (M), and occurrence data for each species was split into training (70%) and evaluation (30%). A total of 10 model replications were implemented through the bootstrapping tool. The medians were used through repetitions as a final estimation of the potential distribution of CL vectors. All the models were converted to binary predictions using the minimum training presence threshold value with an error rate of E = 10%. The threshold selection methods were based on lower threshold values, i.e., with a wider distribution of suitable habitat and close to zero errors of omission.

A partial ROC (Receiver Operating Characteristic) was used to evaluate the performance [31] of the models for the Neutral episode of 2012–2015. This approach potentially allows differential weighting of errors of omission (i.e., false negatives, leaving out actual distribution areas) and commission (i.e., false positives, including unsuitable areas in the prediction) and concentrates attention on the parts of error space most relevant to niche modeling [32].

To estimate changes in altitudinal distribution, after the transferences to the La Niña episode of 2010–2011 (Model 2) and the El Niño episode of 2015–2016 (Model 3), we compared the final binary models to a digital elevation model.

## Relationship between the potential richness of the vectors and CL occurrence in Norte de Santander by episode

The relationship between the potential richness of the vectors and the occurrence of at least one CL case was evaluated only for rural localities, because of the uncertainty about the location where the infection occurred in the cases reported in urban areas and small villages and the low number of cases reported in such areas. This administrative division is based on the national surveillance system.

Similarly, because most of the rural localities (>88%) did not have any CL cases, and most of the localities that had CL cases only reported one case during each episode, the total number of CL cases was not considered for this analysis.

The potential richness was estimated as the sum of the pixels in thresholded models for all four vectors of CL. For each rural locality was calculated the percentage coverage for the richness of the vectors (from zero to four *Lutzomyia* species).

We tested the hypothesis that there is a relationship between the presence of at least one CL case and vector richness in rural localities. The hypothesis was evaluated independently for each episode: La Niña 2010–2011, Neutral 2012–2015 and El Niño 2015–2016. Therefore, a log-binomial regression analysis was conducted to calculate the prevalence ratio using the frequency of the rural localities with at least one CL case as the dependent variable and the percentage of area with a certain richness of *Lutzomyia* species as the predictor variable. The Prevalence Ratio (PR) was reported with 95% confidence intervals (CI 95%). This analysis was conducted using Stata 14 software.

## Ethics statement

This research did not receive IRB approval because the vector information was from database of published records, and the information of CL cases was from surveillance system and it was anonymized.

**Table 1. Results of partial ROC analysis to test the statistical significance of ecological niche model predictions for the Neutral 2012–2015 episode.**

| Species | Mean[a] | Minimum | Maximum | Correct classification rate | Omission error (fraction) |
|---|---|---|---|---|---|
| *Lutzomyia gomezi* | 1.51 | 1.15 | 1.78 | 0.78 | 0.21 |
| *Lutzomyia ovallesi* | 1.50 | 0.85 | 1.88 | 0.73 | 0.26 |
| *Lutzomyia panamensis* | 1.43 | 0.94 | 1.82 | 0.92 | 0.07 |
| *Lutzomyia spinicrassa* | 1.98 | 1.97 | 2 | 1 | 0 |

[a]A value of 1.0 is equivalent to the performance of a random classifier. These results are based on 100 bootstrap replicates.

## Results

The complete occurrence database included 231 records of the presence of vectors, including *L. gomezi* (n = 107), *L. ovallesi* (n = 39), *L. panamensis* (n = 48) and *L. spinicrassa* (n = 37). The statistical assessment of the potential distribution for neutral models (the Neutral 2012–2015 episode) showed a high performance (ROCp > 1.42) (Table 1).

The potential distribution of the four *Lutzomyia* species during the Neutral 2012–2015, the La Niña 2010–2011, and the El Niño 2015–2016 episodes are shown in the Figs 1 to 4. The results suggest that, for *L. ovallesi* and *L. panamensis*, an increase in environmentally suitable area occurred during the El Niño 2015–2016 episode, and therefore, an expansion occurred in its distribution mainly for the Orinoquía and Amazon regions (Fig 1C and Fig 2C). In addition, for three of the four species (*L. gomezi*, *L. ovallesi* and *L. panamensis*), an increase in the range of the altitudinal distribution was found, showing environmentally suitable zones above 1,700 m.a.s.l. During the La Niña 2010–2011 episode, a decrease occurred in the suitable area of occupation for *L. gomezi* in several regions of the country (Fig 3B) and for *L. spinicrassa* (Fig

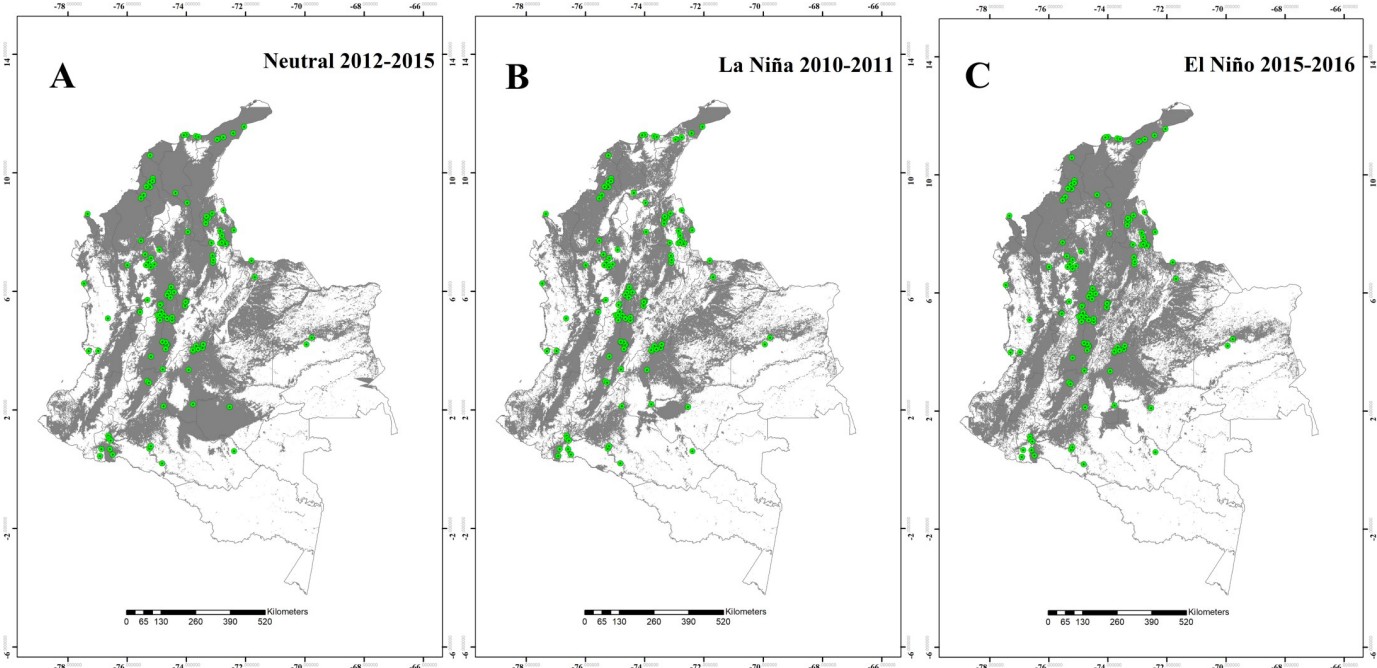

**Fig 1. Potential distribution maps for *Lutzomyia ovallesi*.** (A) Neutral 2012–2015 episode. (B) La Niña 2010–2011 episode. (C) El Niño 2015–2016 episode. Models were calibrated across the hypothesized area of dispersion (M) and transferred across all Colombia. Green points are occurrences, gray areas are modeled suitable conditions, and white areas are unsuitable conditions. Points correspond to data in S1 Appendix, and maps were done using Maxent and ArcGIS software.

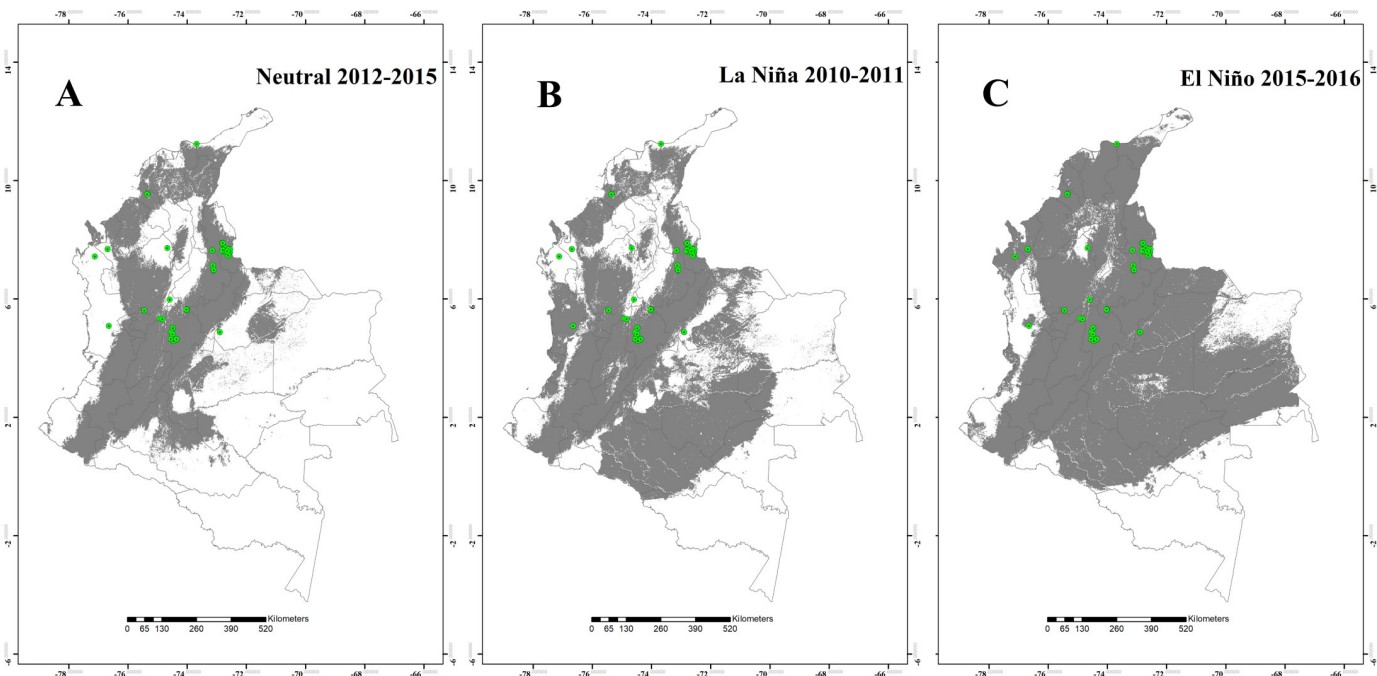

**Fig 2. Potential distribution maps for *Lutzomyia panamensis*.** (A) Neutral 2012–2015 episode. (B) La Niña 2010–2011 episode. (C) El Niño 2015–2016 episode. Models were calibrated across the hypothesized area of dispersion (M) and transferred across all Colombia. Green points are occurrences, gray areas are modeled suitable conditions, and white areas are unsuitable conditions. Points correspond to data in S1 Appendix, and maps were done using Maxent and ArcGIS software.

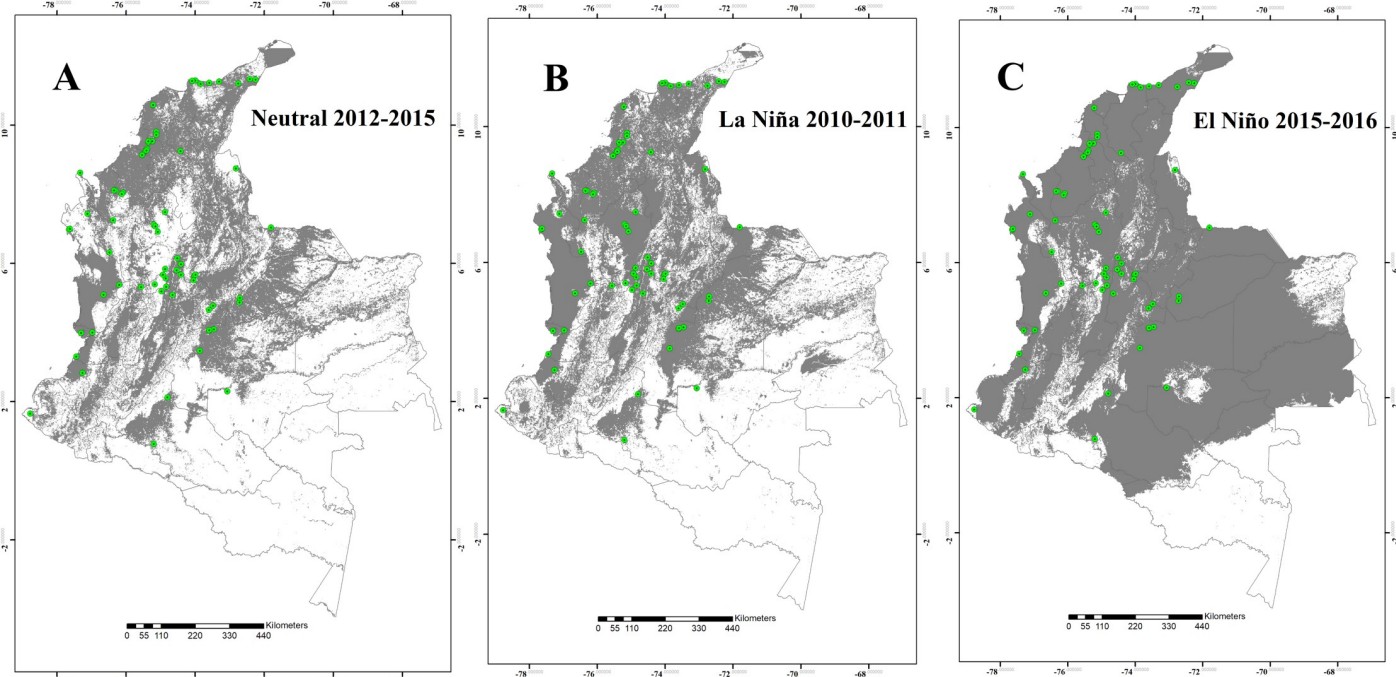

**Fig 3. Potential distribution maps for *Lutzomyia gomezi*.** (A) Neutral 2012–2015 episode. (B) La Niña 2010–2011 episode. (C) El Niño 2015–2016 episode. Models were calibrated across the hypothesized area of dispersion (M) and transferred across all Colombia. Green points are occurrences, gray areas are modeled suitable conditions, and white areas are unsuitable conditions. Points correspond to data in S1 Appendix, and maps were done using Maxent and ArcGIS software.

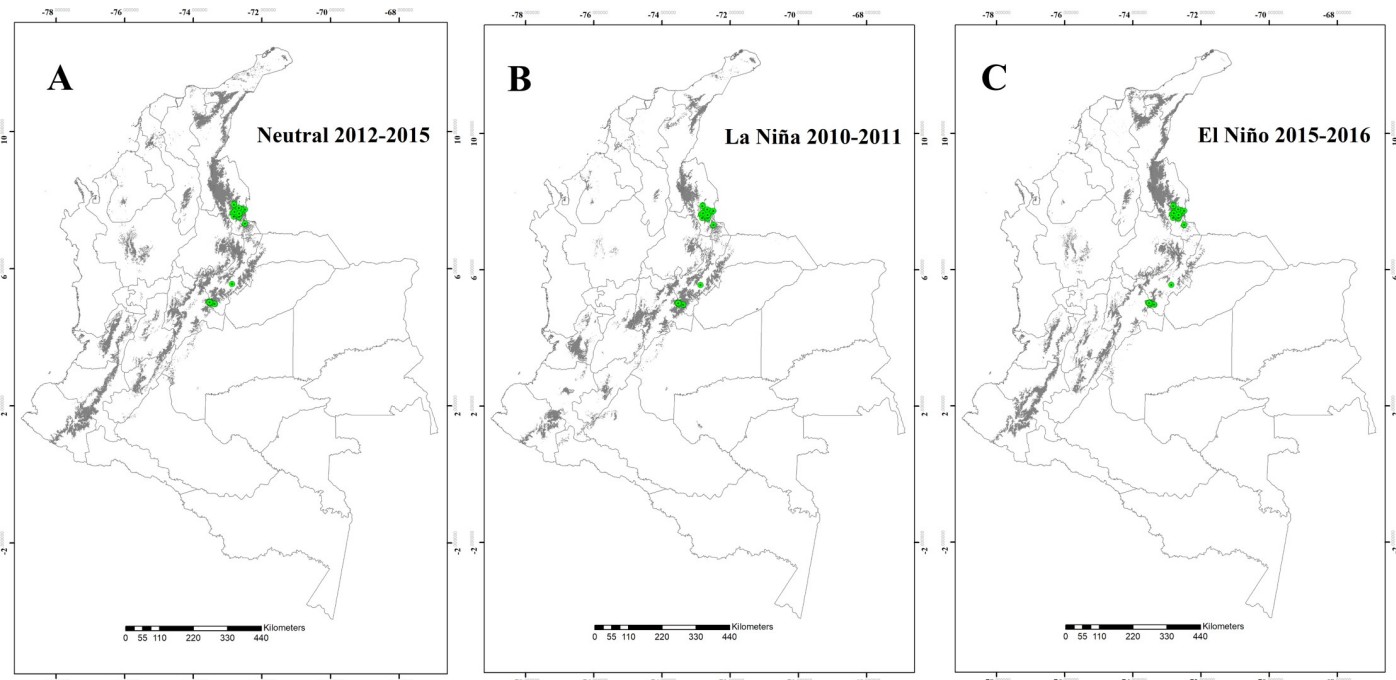

**Fig 4. Potential distribution maps for *Lutzomyia spinicrassa*.** (A) Neutral 2012–2015 episode. (B) La Niña 2010–2011 episode. (C) El Niño 2015–2016 episode. Models were calibrated across the hypothesized area of dispersion (M) and transferred across all Colombia. Green points are occurrences, gray areas are modeled suitable conditions, and white areas are unsuitable conditions. Points correspond to data in S1 Appendix, and maps were done using Maxent and ArcGIS software.

4B) in the Caribbean and Andean region. On the other hand, *L. ovallesi* increased its potential distribution area in this same period, especially in the Orinoco region (Fig 1B).

Most of the zones with no analogous climates (similarity to M = 0) occurred during the El Niño 2015–2016 episode for *L. gomezzi*, *L. ovallesi* and *L. panamensis*, mainly in the north of the country (Fig 5). The northeast of Norte de Santander corresponds to a zone with no analogous climates for *L. gomezzi*, *L. ovallesi* and *L. panamensis* during the El Niño 2015–2016 episode (Fig 5B, 5D and 5F).

## Cases of CL

Of the 1,705 localities identified in Norte de Santander, 40 (2.3%) are urban areas, and 77 (4.5%) are small villages. Among the 1,588 rural localities, during the Neutral episode of 2012–2015 (Fig 6A), 10.8% (172 localities) presented CL cases (372 cases in total, with a median 0 per locality, and ranging from 1 to 24 cases per locality when cases were present); during the La Niña episode of 2010–2011, only 5% (79 localities) had CL cases (131 cases in total, with a median of 0 per locality, and ranging from 1 to 17 cases per locality when cases were present) (Fig 6B); and during El Niño episode of 2015–2016, 12% (191 localities) had CL cases (511 cases in total, with a median of 0 per locality, and ranging from 1 to 19 cases per locality when cases were present) (Fig 6C).

## Relationship between the potential richness and CL occurrence in Norte de Santander by episode

During the El Niño 2015–2016 episode, the occurrence of at least one CL case in rural localities was related to a higher percentage of area with a richness of vectors of CL = 4 (PR 1.012; CI

La Niña 2010-2011     El Niño 2015-2016

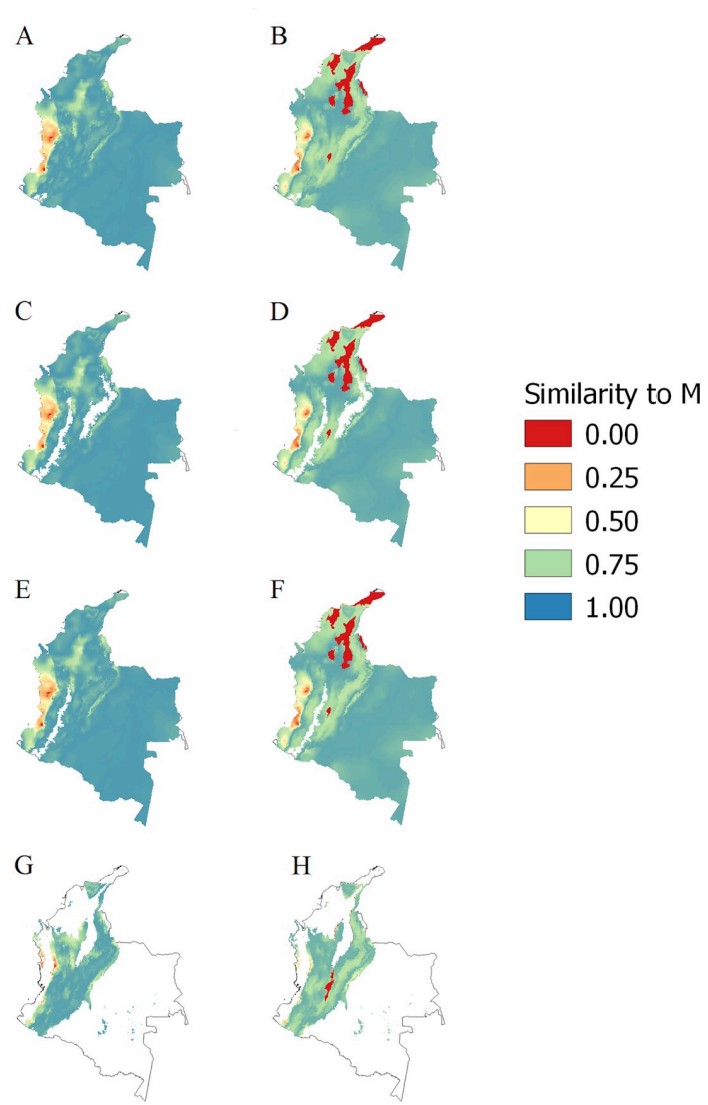

**Fig 5. Climatic similarity estimated by MOP metric between M and transference zone for each *Lutzomyia* vector.**
(A) and (B) *L. gomezzi*. (C) and (D) *L. ovallesi*. (E) and (F) *L. panamensis*. (G) and (H) *L. spinicrassa*. White areas are not part of M. Maps were done using ntbox package from R software.

95% 1.008–1.015, p<0.001). Therefore, for each increment of 1% in area of vector richness = 4 in a rural locality, the frequency of rural localities with at least one CL case increased 1.2% during the El Niño 2015–2016 episode. Similar results were found during the Neutral 2012–2015 episode for richness ≥3 (PR 1.013; CI 95% 1.008–1.018, p<0.001), but no association was found with La Niña 2010–2011 (PR 1.000; CI 95% 0.994–1.006, p = 0.94) (S2 Table).

These statistical findings are related to the observations presented in Fig 7. The intersection between rural localities with potential richness of vectors ≥3 and rural localities with at least one CL case during the Neutral 2012–2015 episode was lower and corresponded to the center zone of the state (Fig 7A). The lowest intersection between rural localities with potential

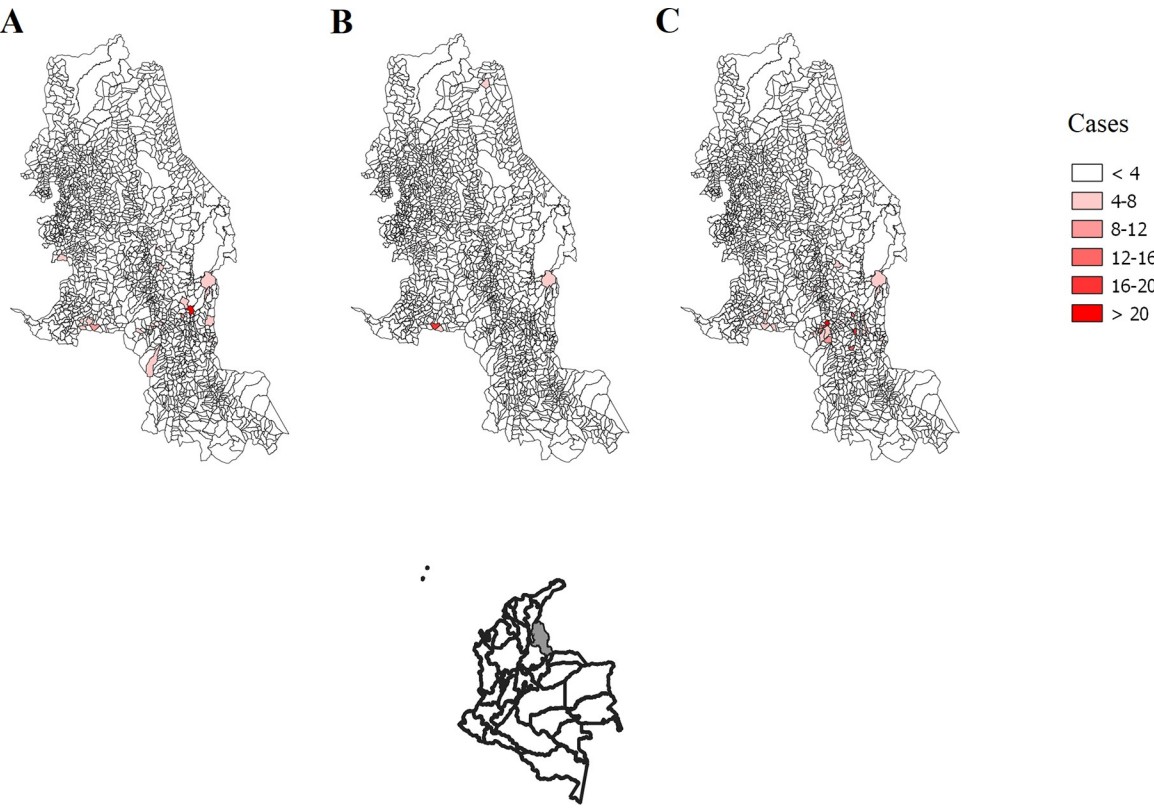

**Fig 6. CL cases in rural localities in Norte de Santander during episodes.** (A) Neutral 2012–2015 episode. (B) La Niña 2010–2011 episode. (C) El Niño 2015–2016 episode. The information was supplied from Laboratorio de Salud Pública of Norte de Santander. Maps were done using QGis software.

richness of vectors ≥ 3 and rural localities with at least one CL case corresponded to the La Niña 2010–2011 episode (Fig 7B). During the El Niño 2015–2016 episode, nearly the entire state of Norte de Santander (except the northeast region) presented a potential richness of

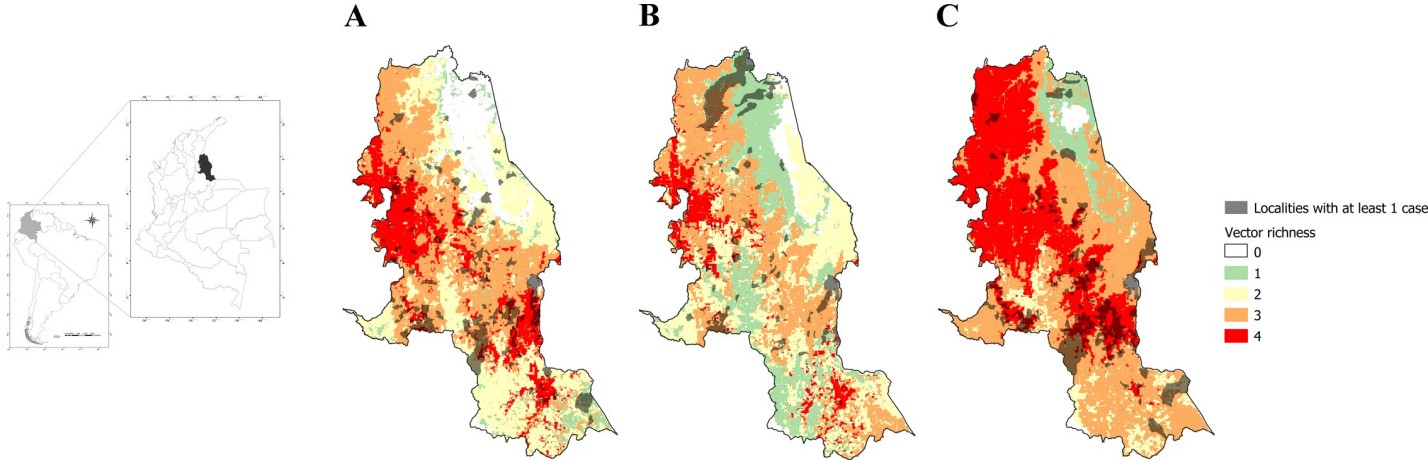

**Fig 7. Intersection of rural localities in Norte de Santander with at least one CL case and potential richness of vectors during episodes.** (A) Neutral 2012–2015 episode. (B) La Niña 2010–2011 episode. (C) El Niño 2015–2016 episode. The information was supplied from Laboratorio de Salud Pública of Norte de Santander. Maps were done using QGis software.

vectors ≥3, and the intersection with rural localities with at least one CL case represented an important extension in the southern and eastern regions of the state (Fig 7C).

## Discussion

Previous works have implemented niche modeling to predict the potential distribution of CL vectors [33–37] and to obtain risk maps of the disease from the co-occurrence (richness) of vectors [38]. The effects of climate change on the predicted distribution of these vectors when niche modeling is implemented also have been analyzed [13,39,40]. However, to our knowledge, this is the first study that implements niche modeling to predict the change in the potential distribution of CL vectors associated with the episodes of the ENSO cycle. Additionally, it is the first to evaluate if these changes in the potential distribution impact the occurrence of cases of CL.

In general, the evaluated vector species tend to distribute in the Andean region; however, *L. panamensis* showed the widest geographical distribution in the three episodes. This result is in accordance with the predicted distribution of this vector in Colombia [7]. In Colombia *L. panamensis* has been reported with anthropophilic activity [41], and transferences from our models in the El Niño 2015–2016 and La Niña 2010–2011 episodes indicated a high percentage of distribution in Colombia, suggesting that this insect could be easily adapted to anthropogenically disturbed environments. The frequent alterations of natural ecosystems generated by human colonization in the Colombian Orinoquía and Amazonia have increased the ecological and environmental conditions that favor the presence and variety of arthropods with importance in public health, including vectors of leishmaniasis [42].

The spatial distributions of insects vectors have been evaluated for most diseases, and particular attention has been given to latitudinal increases that will put populations at risk [43]. However, factors that promote shifts in the altitudinal distribution have received little attention. Our results indicate that the range increased in the altitudinal distribution for three species of vectorial importance; this observation may constitute a warning sign for the health authorities. Elevational shifts also were previously predicted for *L. longipalpis* and *L. evansi*, vectors of visceral leishmaniasis in Colombia, and in certain regions in the Caribbean Coast [43].

In the case of the CL vector species studied here, we assume that the observed elevational shifts in the distribution of the species may have been induced by El Niño 2015–2016 and La Niña 2010–2011 episodes. However, these changes were probably mediated by the land cover and the suitability of the habitat for the establishment of viable populations of vectors, particularly in forests [6,44] and perennial crops (e.g., coffee and cocoa) [5,45,46].

An increase in temperature has an impact on the life cycles of CL vectors and *Leishmania* parasites. For example, an increase of 4˚C (from 20˚C to 24˚C) can reduce the egg-to-adult development time of *L. anthophora* to 33.8 days [47]. For this same vector, an increase in temperature from 24˚C to 28˚C had no apparent effect on egg production, but egg production was greatly reduced in adults kept at 32˚C compared to those kept at 24 and 28˚C [47]. Similarly, in a culture test with *Leishmania brazilensis*, a significant increase in infectivity was observed when promastigotes were transferred from 26˚C to 34˚C, and they changed morphologically to resemble intracellular amastigotes [48].

Our results assume an equal velocity of expansion during each episode to new suitable environments for the four vector species. However, previous studies have shown that for example: one genotype of *L. longipalpis* expanded faster than another in new environments [49], making it possible to infer that the velocity of colonization of new suitable environments induced by the ENSO cycle occurred with different velocity for each vector of CL.

We recognize that the probability of new cases occurring may be reduced in certain periods during La Niña episodes, particularly when heavy rainfall events occur that hamper access to remote and sylvatic areas by lumberjacks and other groups of people who may be occupationally exposed to the disease. Such extreme weather conditions can act as secondary drivers, not associated with the biology of CL contagion, yet reducing the occurrence of cases; this possibility affects the interpretation of our results.

For future studies, we suggest the use of remote sensors of climate variables, with the aim of obtaining a complete dataset for the whole country, and modeling more precisely the change in the potential distribution of CL vectors in association with anomalies in the rainfall and temperature induced by the El Niño and La Niña episodes. This last recommendation is made because the temperature and rainfall data for Colombia provided by IDEAM included 1,998 pluviometric and 519 temperature stations, but only 74% of the municipalities had at least one pluviometric station, and 33% of the municipalities had at least one temperature station. Missing data between 2007 and 2016 inside the pluviometric stations corresponded to 20% and 32% in the case of the temperature stations.

One of limitations of this study was that each case was georeferenced to the locality of the patient's residence. This was because information on the possible place of transmission was not available in the surveillance data. However, it is likely that the transmission occurred in the same area, given that only the rural cases were included in this report. Likewise, it was difficult to establish the date when the transmission occurred, and the symptom onset date was not available, so the date when the case was recorded by surveillance system was used. Another related limitation was that the population of each rural locality was unavailable. For this reason, the incidence rate and the incidence rate ratio could not be estimated. Additionally, data on migratory movements were not available for rural localities. Also, considering than in Colombia it has been reported the circulation of at least six *Leishmania* species in CL cases, the absence of identification of species in the CL cases limited the analysis about the relation between the vector and the parasite. Similarly, the presence of animal reservoirs was not considered in this study; this topic could serve as a focus for future research. Furthermore, the MOP metric showed the existence of no analogous climates in the northeast of Norte de Santander for some of the vector species evaluated, which lead to make a cautious interpretation of the estimation of richness of vectors in this zone. For last, we recognize gaps in the prediction of potential distribution of CL vectors, based on incomplete data; publication, taxonomic, and misdetermination bias; and not well distributed data in the country.

In conclusion, CL is a complex disease associated with climate determinants. Our results show the influence of the most extreme and longest episodes of the ENSO cycle on the potential distribution of CL vectors and the occurrence of the disease. The anomalies in rainfall and temperature induced by the episodes—La Niña 2010–2011, Neutral 2012–2015 and El Niño 2015–2016—produced changes in the potential distribution and richness of the CL vectors in Colombia due to an increase or reduction in the environmentally suitable area. In rural localities of Norte de Santander during the Neutral 2012–2015 and El Niño 2015–2016 episodes, the occurrence of at least one CL case was related to a higher percentage of area with greater richness of vectors. The present study sheds light on the importance of the ENSO cycle in the dynamics of the disease and the necessity for monitoring the climate variability to improve the early attention to CL outbreaks in the country.

## Supporting information

**S1 Appendix. Presence records of *Lutzomyia* species in Colombia.**
(XLSX)

**S1 Table. Episodes of the ENSO cycle between 2007 and 2016 according to the ONI index values of the National Oceanic and Atmospheric Administration.**
(DOCX)

**S2 Table. Relationship between occurrence of at least one CL case in each locality and the percentage of local area with a richness of vectors, Norte de Santander (n = 1,588 localities)***.**
(DOCX)

**S1 Fig. Uncertainty of the records without specific coordinates used in the potential distribution models.**
(TIF)

## Author Contributions

**Conceptualization:** Mariano Altamiranda-Saavedra, Juan David Gutiérrez, Ruth A. Martínez-Vega.

**Data curation:** Mariano Altamiranda-Saavedra, Juan David Gutiérrez, Astrid Araque, Juan David Valencia-Mazo, Reinaldo Gutiérrez, Ruth A. Martínez-Vega.

**Formal analysis:** Mariano Altamiranda-Saavedra, Juan David Gutiérrez, Astrid Araque, Juan David Valencia-Mazo, Reinaldo Gutiérrez, Ruth A. Martínez-Vega.

**Methodology:** Mariano Altamiranda-Saavedra, Juan David Gutiérrez, Ruth A. Martínez-Vega.

**Resources:** Astrid Araque.

**Writing – original draft:** Mariano Altamiranda-Saavedra, Juan David Gutiérrez, Ruth A. Martínez-Vega.

**Writing – review & editing:** Astrid Araque, Juan David Valencia-Mazo, Reinaldo Gutiérrez, Ruth A. Martínez-Vega.

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
