## [Decision Letter · Decision Letter 0]

19 Feb 2020

Dear Dr. Martínez-Vega,

Thank you very much for submitting your manuscript "Effect of El Niño Southern Oscillation cycle on the potential distribution of cutaneous leishmaniasis vector species in Colombia" for consideration at PLOS Neglected Tropical Diseases. As with all papers reviewed by the journal, your manuscript was reviewed by members of the editorial board and by several independent reviewers. In light of the reviews (below this email), we would like to invite the resubmission of a significantly-revised version that takes into account the reviewers' comments. 

We cannot make any decision about publication until we have seen the revised manuscript and your response to the reviewers' comments. Your revised manuscript is also likely to be sent to reviewers for further evaluation.

Sincerely,

Hans-Peter Fuehrer

Deputy Editor

Reviewer's Responses to Questions

**Key Review Criteria Required for Acceptance?**

**Methods**

-Are the objectives of the study clearly articulated with a clear testable hypothesis stated?

-Is the study design appropriate to address the stated objectives?

-Is the population clearly described and appropriate for the hypothesis being tested?

-Is the sample size sufficient to ensure adequate power to address the hypothesis being tested?

-Were correct statistical analysis used to support conclusions?

-Are there concerns about ethical or regulatory requirements being met?

Reviewer #1: Material and methods were properly described and used to achieve the main goals of the study. The limitations of the data were also critically addressed in Discussion section, without belittle the findings.

Reviewer #2: Please see Summary and General Comments

Reviewer #3: The objectives are related to the hypothesis raised.

The approaches used to resolve data inconsistencies or lack of resolution are appropriate.

The sample size is sufficient for the analysis carried out. The variables are relevant but in the case of the land cover layer they use in the different models proposed but nothing is said about it in the Results and discussion section.

**Results**

-Does the analysis presented match the analysis plan?

-Are the results clearly and completely presented?

-Are the figures (Tables, Images) of sufficient quality for clarity?

Reviewer #1: The data analyses showed the effect of climate changes caused by El Nino in distibution of four vector species of cutaneuous leishmaniasis in Colombia, which are completely new. The Results section is very descriptive, which is a consequence from the type of study. Minor changes in text and in layout of figures should be considered (details are in comments in the attached pdf).

Reviewer #2: Please see Summary and General Comments

Reviewer #3: The analyses are adequate and the results are quite clear. The figures are adequate and sufficient.

It is recommended that the explanation in the text of the Figures in general should be done in the order in which they appear (A, B, C) and/or be consistent with the models are presented in Materials and Methods: neutral episode, "La Niña" episode and finally "El Niño" episode.

**Conclusions**

-Are the conclusions supported by the data presented?

-Are the limitations of analysis clearly described?

-Do the authors discuss how these data can be helpful to advance our understanding of the topic under study?

-Is public health relevance addressed?

Reviewer #1: The findings is worthy to be published. The discussion section, as well as the Introduction section, are well written and substantiated. The limitations and necessary advances about the study were pointed out.

Reviewer #2: Please see Summary and General Comments

Reviewer #3: The limitations of the work are well delimited and they made relevant approaches to solve them losing spatial resolution but that served to mark a trend.

The main problem in the areas affected by CL is the recording of the case because of the "time span" until the symptoms appear, which makes it difficult to get the date and probable place of infection of this pathology.

**Editorial and Data Presentation Modifications?**

Reviewer #1: Minor changes are indicated throughout the manuscript file (see attached pdf file).

Reviewer #2: Please see Summary and General Comments

Reviewer #3: “Minor Revision”

**Summary and General Comments**

Reviewer #1: There is no doubt about how weather conditions affect vector-borne diseases, but it is necessary to study the true impact of the recent climate changes in burden of vector-borne diseases and the authors have collaborated on that, analysing the climate effect in distribution of Lutzomyia vectors on cutaneuous leishmaniasis cases. They report the analyses based on recent El Nino - La Nina cycle, which is a short period, but is worthy to be published, because I considere that small data collection is value to further being part of larger and comprehensive studies.

Reviewer #2: This well-written manuscript presents a model for the effect of El Niño Southern Oscillation (ENSO) on the distribution of vectors and cases of cutaneous leishmaniasis in an endemic region of Colombia. There are some general concerns.

1) Introduction: it is not exactly right that "The effect of the ENSO cycle over the predicted distribution map of the vectors of CL has not yet been assessed, nor has any association of this effect with changes in the occurrence of cases of CL been assessed." (lines 149-151). See, for instance, Yamada et al. (2016, Parasite Epidemiol Control); Ferreira de Souza et al. (2015, Geospat Health); Chaves et al. (2014, PLoS Negl Trop Dis) & Chaves et al. (2008, PLoS Negl Trop Dis).

2) Methods: I am not sure why you have to choose just one type of each episode (Neutral, La Niña, El Niño) since you have other strong and long events in the period. Their inclusion should add information to the analysis. In any case, the justification that they "corresponded to the most extreme events with the longest duration of their type" (lines 219-220) is not exactly correct because the first La Niña strong episode (2007-2008) is longer (14, not 13 months) than the one used in the analyses (La Niña 2010-2011, 13 months). 

3)Methods: please provide some uncertainty measure for pROC (e.g., minimum and maximum based on bootstrapping). The problem of partial ROC analysis to test statistical significance of ecological niche model predictions is that it does not provide a measure of how much good is the prediction, you just know that 1.0 corresponds to a random classifier, so you can test significance but this information which is too vague to assess the quality of the prediction. Even knowing the limitations of the regular AUC analysis in such a setting, I would like to see the % of correct classifications for presence points and for a random sample of absence points, and also the number of wrong presence predictions. 

4) Methods: The authors used a random sample of 30% of the distribution data to evaluate the model. Please make sure to give a detailed description of this process. Is the 30% used to test and validation was done in the other 70%? Did you use some cross-validation? Is the error greater or lower for points located in the state of Norte de Santander as compared to points outside?

5) Discussion: should highlight problems in prediction based on incomplete data (only those that could be assessed -- publication bias; mainly presence data - misclassification bias; data not well distributed in the area, etc.).

Reviewer #3: In general the study is well supported by the data it uses and provides in the supplements. Some minor revisions and further details are highlighted and requested in the manuscript.

PLOS authors have the option to publish the peer review history of their article (what does this mean?). If published, this will include your full peer review and any attached files.

Reviewer #1: No

Reviewer #2: No

Reviewer #3: No
---

## [Decision Letter · Decision Letter 1]

24 Apr 2020

Dear Dr. Martínez-Vega,

We are pleased to inform you that your manuscript 'Effect of El Niño Southern Oscillation cycle on the potential distribution of cutaneous leishmaniasis vector species in Colombia' has been provisionally accepted for publication in PLOS Neglected Tropical Diseases.

Best regards,

Hans-Peter Fuehrer

Deputy Editor

Hans-Peter Fuehrer

Deputy Editor

Reviewer's Responses to Questions

**Key Review Criteria Required for Acceptance?**

**Methods**

-Are the objectives of the study clearly articulated with a clear testable hypothesis stated?

-Is the study design appropriate to address the stated objectives?

-Is the population clearly described and appropriate for the hypothesis being tested?

-Is the sample size sufficient to ensure adequate power to address the hypothesis being tested?

-Were correct statistical analysis used to support conclusions?

-Are there concerns about ethical or regulatory requirements being met?

Reviewer #1: (No Response)

Reviewer #2: I have no further concerns regarding the methods.

Reviewer #3: The modifications and/or suggestions made were taken into account, and were incorporated into the manuscript .

**Results**

-Does the analysis presented match the analysis plan?

-Are the results clearly and completely presented?

-Are the figures (Tables, Images) of sufficient quality for clarity?

Reviewer #1: (No Response)

Reviewer #2: I have no further concerns regarding the results.

Reviewer #3: The results are clearly presented from the changes made.

**Conclusions**

-Are the conclusions supported by the data presented?

-Are the limitations of analysis clearly described?

-Do the authors discuss how these data can be helpful to advance our understanding of the topic under study?

-Is public health relevance addressed?

Reviewer #1: (No Response)

Reviewer #2: I have no further concerns regarding the conclusions.

Reviewer #3: The discussion and conclusions were reordered and improved.

**Editorial and Data Presentation Modifications?**

Reviewer #1: (No Response)

Reviewer #2: The authors accepted all my recommendations and modified the manuscript accordingly. However, I think the manuscript should undergo an English language revision. For instance, the new text included in line 150 needs revision (maybe "no previous study simultaneously appraised"; since the verb is in this sentence, in item (b) you do not need another verb ("evaluate"). Also, check text in line 535 ("which *lead to make* a cautious interpretation"). There are also some wordy sentences and some incorrect use of commas.

Reviewer #3: The manuscript is considered ready for publication considering before final acceptance the adequacy of the format/style of PNTD.

**Summary and General Comments**

Reviewer #1: The authors addressed the key points raised during revision. No further comments from this reviewer.

Reviewer #2: The authors accepted all my recommendations and modified the manuscript accordingly. However, I think the manuscript should undergo an English language revision. For instance, the new text included in line 150 needs revision (maybe "no previous study simultaneously appraised", since the verb is in this sentence, in item (b) you do not need another verb ("evaluate"). Also, check text in line 535 ("which *lead to make* a cautious interpretation"). There are also some wordy sentences and some incorrect use of commas.

Reviewer #3: -

PLOS authors have the option to publish the peer review history of their article (what does this mean?). If published, this will include your full peer review and any attached files.

Reviewer #1: No

Reviewer #2: Yes: Guilherme Loureiro Werneck

Reviewer #3: No

---

## [Editor Report · Acceptance letter]

15 May 2020

Dear Dr. Martínez-Vega,

We are delighted to inform you that your manuscript, "Effect of El Niño Southern Oscillation cycle on the potential distribution of cutaneous leishmaniasis vector species in Colombia," has been formally accepted for publication in PLOS Neglected Tropical Diseases.

Best regards,

Serap Aksoy

Editor-in-Chief

Shaden Kamhawi

Editor-in-Chief
